# Episodic Memory in Lifelong Language Learning

**Cyprien de Masson d'Autume, Sebastian Ruder, Lingpeng Kong, Dani Yogatama**
DeepMind
London, United Kingdom
`{cyprien,ruder,lingpenk,dyogatama}@google.com`

## Abstract

We introduce a lifelong language learning setup where a model needs to learn from a stream of text examples without any dataset identifier. We propose an episodic memory model that performs sparse experience replay and local adaptation to mitigate catastrophic forgetting in this setup. Experiments on text classification and question answering demonstrate the complementary benefits of sparse experience replay and local adaptation to allow the model to continuously learn from new datasets. We also show that the space complexity of the episodic memory module can be reduced significantly ($\sim$50-90%) by randomly choosing which examples to store in memory with a minimal decrease in performance. We consider an episodic memory component as a crucial building block of general linguistic intelligence and see our model as a first step in that direction.

## 1 Introduction

The ability to continuously learn and accumulate knowledge throughout a lifetime and reuse it effectively to adapt to a new problem quickly is a hallmark of general intelligence. State-of-the-art machine learning models work well on a single dataset given enough training examples, but they often fail to isolate and reuse previously acquired knowledge when the data distribution shifts (e.g., when presented with a new dataset)—a phenomenon known as *catastrophic forgetting* (McCloskey & Cohen, 1989; Ratcliff, 1990).

The three main approaches to address catastrophic forgetting are based on: (i) augmenting the loss function that is being minimized during training with extra terms (e.g., a regularization term, an optimization constraint) to prevent model parameters learned on a new dataset from significantly deviating from parameters learned on previously seen datasets (Kirkpatrick et al., 2017; Zenke et al., 2017; Chaudhry et al., 2018), (ii) adding extra learning phases such as a knowledge distillation phase, an experience replay (Schwarz et al., 2018; Wang et al., 2019), and (iii) augmenting the model with an episodic memory module (Sprechmann et al., 2018). Recent methods have shown that these approaches can be combined—e.g., by defining optimization constraints using samples from the episodic memory (Lopez-Paz & Ranzato, 2017; Chaudhry et al., 2019).

In language learning, progress in unsupervised pretraining (Peters et al., 2018; Howard & Ruder, 2018; Devlin et al., 2018) has driven advances in many language understanding tasks (Kitaev & Klein, 2018; Lee et al., 2018). However, these models have been shown to require a lot of in-domain training examples, rapidly overfit to particular datasets, and are prone to catastrophic forgetting (Yogatama et al., 2019), making them unsuitable as a model of general linguistic intelligence.

In this paper, we investigate the role of episodic memory for learning a model of language in a lifelong setup. We propose to use such a component for *sparse experience replay* and *local adaptation* to allow the model to continually learn from examples drawn from different data distributions. In experience replay, we randomly select examples from memory to retrain on. Our model only performs experience replay very sparsely to consolidate newly acquired knowledge with existing knowledge in the memory

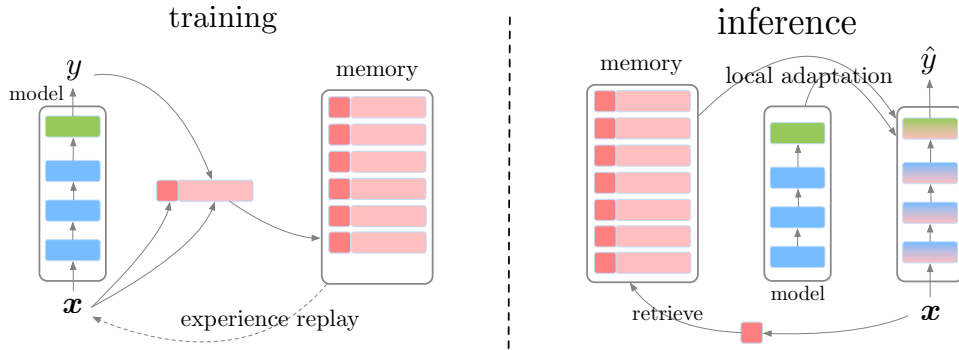

Figure 1: An illustration of our model and how it interacts with the key-value memory module during training (left) and inference (right). During training, newly seen examples are used to update the base model and stored in the memory. At certain intervals, we sample examples from the memory and perform gradient updates on the base model (experience replay). During inference, we retrieve examples whose keys are similar to a test example under consideration to fine-tune the model (local adaptation). We use the fine-tuned model to make a prediction and then discard it—keeping the base model for other predictions.

into the model. We show that a 1% experience replay to learning new examples ratio is sufficient. Such a process bears some similarity to memory consolidation in human learning (McGaugh, 2000). In local adaptation, we follow Memory-based Parameter Adaptation (MbPA; Sprechmann et al., 2018) and use examples retrieved from memory to update model parameters used to make a prediction of a particular test example.

Our setup is different from a typical lifelong learning setup. We assume that the model only makes one pass over the training examples, similar to Chaudhry et al. (2019). However, we also assume neither our training nor test examples have dataset identifying information (e.g., a dataset identity, a dataset descriptor). We argue that our lifelong language learning setup—where a model is presented with a stream of examples without an explicit identifier about which dataset (distribution) the examples come from—is a realistic setup to learn a general linguistic intelligence model.[1] Our experiments focus on lifelong language learning on two tasks—text classification and question answering.[2]

Our main contributions in this paper are:

- We introduce a lifelong language learning setup where the model needs to learn from a stream of examples from many datasets (presented sequentially) in one pass, and no dataset boundary or dataset identity is given to the model.

- We present an episodic memory model (§2) that augments an encoder-decoder model with a memory module. Our memory is a key-value memory that stores previously seen examples for sparse experience replay and local adaptation.

- We leverage progress in unsupervised pretraining to obtain good memory key representations and discuss strategies to manage the space complexity of the memory module.

- We compare our proposed method to baseline and state-of-the-art continual learning methods and demonstrate its efficacy on text classification and question answering tasks (§4).

## 2   Model

We consider a continual (lifelong) learning setup where a model needs to learn from a stream of training examples $\{\boldsymbol{x}_t, y_t\}_{t=1}^T$. We assume that all our training examples in the series come from multiple datasets of the same task (e.g., a text classification task, a question answering task), and each dataset comes one after the other. Since all examples come from the same task, the same model can be used to make predictions on all examples. A crucial difference between our continual learning

setup and previous work is that we do not assume that each example comes with a dataset descriptor (e.g., a dataset identity). As a result, the model does not know which dataset an example comes from and when a dataset boundary has been crossed during training. The goal of learning is to find parameters $\mathbf{W}$ that minimize the negative log probability of training examples under our model:

$$\mathcal{L}(\mathbf{W}) = -\sum_{t=1}^{T} \log p(y_t \mid \boldsymbol{x}_t; \mathbf{W}).$$

Our model consists of three main components: (i) an example encoder, (ii) a task decoder, and (iii) an episodic memory module. Figure 1 shows an illustration of our complete model. We describe each component in detail in the following.

## 2.1 Example Encoder

Our encoder is based on the Transformer architecture (Vaswani et al., 2017). We use the state-of-the-art text encoder BERT (Devlin et al., 2018) to encode our input $\boldsymbol{x}_t$. BERT is a large Transformer pretrained on a large unlabeled corpus on two unsupervised tasks—masked language modeling and next sentence prediction. Other architectures such as recurrent neural networks or convolutional neural networks can also be used as the example encoder.

In text classification, $\boldsymbol{x}_t$ is a document to be classified; BERT produces a vector representation of each token in $\boldsymbol{x}_t$, which includes a special beginning-of-document symbol CLS as $x_{t,0}$. In question answering, $\boldsymbol{x}_t$ is a concatenation of a context paragraph $\boldsymbol{x}_t^{\text{context}}$ and a question $\boldsymbol{x}_t^{\text{question}}$ separated by a special separator symbol SEP.

## 2.2 Task Decoder

In text classification, following the original BERT model, we take the representation of the first token $x_{t,0}$ from BERT (i.e., the special beginning-of-document symbol) and add a linear transformation and a softmax layer to predict the class of $\boldsymbol{x}_t$.

$$p(y_t = c \mid \boldsymbol{x}_t) = \frac{\exp(\mathbf{w}_c^\top \mathbf{x}_{t,0})}{\sum_{y \in \mathcal{Y}} \exp(\mathbf{w}_y^\top \mathbf{x}_{t,0})}$$

Note that since there is no dataset descriptor provided to our model, this decoder is used to predict all classes in all datasets, which we assume to be known in advance.

For question answering, our decoder predicts an answer span—the start and end indices of the correct answer in the context. Denote the length of the context paragraph by $M$, and $\boldsymbol{x}_t^{\text{context}} = \{x_{t,0}^{\text{context}}, \dots, x_{t,M}^{\text{context}}\}$. Denote the encoded representation of the $m$-th token in the context by $\mathbf{x}_{t,m}^{\text{context}}$. Our decoder has two sets of parameters: $\mathbf{w}_{\text{start}}$ and $\mathbf{w}_{\text{end}}$. The probability of each context token being the start of the answer is computed as:

$$p(\texttt{start} = x_{t,m}^{\text{context}} \mid \boldsymbol{x}_t) = \frac{\exp(\mathbf{w}_{\text{start}}^\top \mathbf{x}_{t,m}^{\text{context}})}{\sum_{n=0}^{M} \exp(\mathbf{w}_{\text{start}}^\top \mathbf{x}_{t,n}^{\text{context}})}.$$

We compute the probability of the end index of the answer analogously using $\mathbf{w}_{\text{end}}$. The predicted answer is the span with the highest probability after multiplying the start and end probabilities. We take into account that the start index of an answer needs to precede its end index by setting the probabilities of invalid spans to zero.

## 2.3 Episodic Memory

Our model is augmented with an episodic memory module that stores previously seen examples throughout its lifetime. The episodic memory module is used for sparse experience replay and local adaptation to prevent catastrophic forgetting and encourage positive transfer. We first describe the architecture of our episodic memory module, before discussing how it is used at training and inference (prediction) time in §3.

The module is a key-value memory block. We obtain the key representation of $\boldsymbol{x}_t$ (denoted by $\mathbf{u}_t$) using a key network—which is a pretrained BERT model separate from the example encoder. We

freeze the key network to prevent key representations from drifting as data distribution changes (i.e. the problem that the key of a test example tends to be closer to keys of recently stored examples).

For text classification, our key is an encoded representation of the first token of the document to be classified, so $\mathbf{u}_t = \mathbf{x}_{t,0}$ (i.e., the special beginning-of-document symbol). For question answering, we first take the question part of the input $\boldsymbol{x}_t^{\text{question}}$. We encode it using the key network and take the first token as the key vector $\mathbf{u}_t = x_{t,0}^{\text{question}}$.[3] For both tasks, we store the input and the label $\langle \boldsymbol{x}_t, y_t \rangle$ as its associated memory value.

**Write.** If we assume that the model has unlimited capacity, we can write all training examples into the memory. However, this assumption is unrealistic in practice. We explore a simple writing strategy that relaxes this constraint based on random write. In random write, we randomly decide whether to write a newly seen example into the memory with some probability. We find that this is a strong baseline that outperforms other simple methods based on surprisal (Ramalho & Garnelo, 2019) and the concept of forgettable examples (Toneva et al., 2019) in our preliminary experiments. We leave investigations of more sophisticated selection methods to future work.

**Read.** Our memory has two retrieval mechanisms: (i) random sampling and (ii) $K$-nearest neighbors. We use random sampling to perform sparse experience replay and $K$-nearest neighbors for local adaptation, which are described in §3 below.

## 3 Training and Inference

Algorithm 1 and Algorithm 2 outline our overall training and inference procedures.

**Sparse experience replay.** At a certain interval throughout the learning period, we uniformly sample from stored examples in the memory and perform gradient updates of the encoder-decoder network based on the retrieved examples. Allowing the model to perform experience replay at every timestep would transform the problem of continual learning into multitask learning. While such a method will protect the model from catastrophic forgetting, it is expensive and defeats the purpose of a lifelong learning setup. Our experience replay procedure is designed to be performed very sparsely. In practice, we randomly retrieve 100 examples every 10,000 new examples. Note that similar to the base training procedure, we only perform one gradient update for the 100 retrieved examples.

**Local adaptation.** At inference time, given a test example, we use the key network to obtain a query vector of the test example and query the memory to retrieve $K$ nearest neighbors using the Euclidean distance function. We use these $K$ examples to perform local adaptation, similar to Memory-based Parameter Adaptation (Sprechmann et al., 2018). Denote the $K$ examples retrieved for the $i$-th test example by $\{\boldsymbol{x}_i^k, y_i^k\}_{k=1}^K$. We perform gradient-based local adaptation to update parameters of the encoder-decoder model—denoted by $\mathbf{W}$—to obtain local parameters $\mathbf{W}_i$ to be used for the current prediction as follows:[4]

$$\mathbf{W}_i = \underset{\tilde{\mathbf{W}}}{\arg\min} \, \lambda \|\tilde{\mathbf{W}} - \mathbf{W}\|_2^2 - \sum_{k=1}^K \alpha_k \log p(y_i^k \mid \boldsymbol{x}_i^k; \tilde{\mathbf{W}}), \quad (1)$$

where $\lambda$ is a hyperparameter, $\alpha_k$ is the weight of the $k$-th retrieved example and $\sum_{k=1}^K \alpha_k = 1$. In our experiments, we assume that all $K$ retrieved examples are equally important regardless of their distance to the query vector and set $\alpha_k = \frac{1}{K}$. Intuitively, the above procedure locally adapts parameters of the encoder-decoder network to be better at predicting retrieved examples from the memory (as defined by having a higher probability of predicting $y_i^k$), while keeping it close to the base parameters $\mathbf{W}$. Note that $\mathbf{W}_i$ is only used to make a prediction for the $i$-th example, and the

parameters are reset to $\mathbf{W}$ afterwards. In practice, we only perform $L$ local adaptation gradient steps instead of finding the true minimum of Eq. 1.

---

**Algorithm 1** Training

---

**Input:** training examples $\langle \boldsymbol{x}_t, y_t \rangle_{t=1}^{T}$, replay interval $R$
**Output:** parameters $\mathbf{W}$, memory $\boldsymbol{M}$
**for** $t = 1$ **to** $T$ **do**
    **if** $t \bmod R = 0$ **then**
        Sample $S$ examples from $\boldsymbol{M}$.
        Perform gradient updates on $\mathbf{W}$. {experience replay}
    **end if**
    Receive a training example $\langle \boldsymbol{x}_t, y_t \rangle$.
    Perform a gradient update on $\mathbf{W}$ to minimize $-\log p(y_t \mid \boldsymbol{x}_t; \mathbf{W})$.
    **if** store example **then**
        Write $\langle \boldsymbol{x}_t, y_t \rangle$ to memory $\boldsymbol{M}$.
    **end if**
**end for**

---

**Algorithm 2** Inference

---

**Input:** test example $\boldsymbol{x}_i$, parameters $\mathbf{W}$, memory $\boldsymbol{M}$
**Output:** test prediction $\hat{y}_i$
Compute query representation $\mathbf{u}_i$ from $\boldsymbol{x}_i$.
Find $K$ nearest neighbors of $\mathbf{u}_i$ from $\boldsymbol{M}$.
$\mathbf{W}_i \leftarrow \mathbf{W}$
**for** $l = 1$ **to** $L$ **do**
    Perform a gradient update on $\mathbf{W}_i$ to minimize Eq. 1. {local adaptation}
**end for**
$\hat{y}_i = \arg\max_y p(y \mid \boldsymbol{x}_i; \mathbf{W}_i)$

---

## 4 Experiments

In this section, we evaluate our proposed model against several baselines on text classification and question answering tasks.

### 4.1 Datasets

**Text classification.** We use publicly available text classification datasets from Zhang et al. (2015) to evaluate our models (`http://goo.gl/JyCnZq`). This collection of datasets includes text classification datasets from diverse domains such as news classification (AGNews), sentiment analysis (Yelp, Amazon), Wikipedia article classification (DBPedia), and questions and answers categorization (Yahoo). Specifically, we use AGNews (4 classes), Yelp (5 classes), DBPedia (14 classes), Amazon (5 classes), and Yahoo (10 classes) datasets. Since classes for Yelp and Amazon datasets have similar semantics (product ratings), we merge the classes for these two datasets. In total, we have 33 classes in our experiments. These datasets have varying sizes. For example, AGNews is ten times smaller than Yahoo. We create a balanced version all datasets used in our experiments by randomly sampling 115,000 training examples and 7,600 test examples from all datasets (i.e., the size of the smallest training and test sets). We leave investigations of lifelong learning in unbalanced datasets to future work. In total, we have 575,000 training examples and 38,000 test examples.

**Question answering.** We use three question answering datasets: SQuAD 1.1 (Rajpurkar et al., 2016), TriviaQA (Joshi et al., 2017), and QuAC (Choi et al., 2018). These datasets have different characteristics. SQuAD is a reading comprehension dataset constructed from Wikipedia articles. It includes almost 90,000 training examples and 10,000 validation examples. TriviaQA is a dataset with question-answer pairs written by trivia enthusiasts and evidence collected retrospectively from Wikipedia and the Web. There are two sections of TriviaQA, Web and Wikipedia, which we treat as separate datasets. The Web section contains 76,000 training examples and 10,000 (unverified) validation examples, whereas the Wikipedia section has about 60,000 training examples and 8,000 validation examples. QuAC is an information-seeking dialog-style dataset where a student asks questions about a Wikipedia article and a teacher answers with a short excerpt from the article. It has 80,000 training examples and approximately 7,000 validation examples.

### 4.2 Models

We compare the following models in our experiments:

- ENC-DEC: a standard encoder-decoder model without any episodic memory module.

- A-GEM (Chaudhry et al., 2019): Average Gradient Episodic Memory model that defines constraints on the gradients that are used to update model parameters based on retrieved examples from the memory. In its original formulation, A-GEM requires dataset identifiers and randomly samples examples from previous datasets. We generalize it to the setting without dataset identities by randomly sampling from the episodic memory module at fixed intervals, similar to our method.
- REPLAY: a model that uses stored examples for sparse experience replay without local adaptation. We perform experience replay by sampling 100 examples from the memory and perform a gradient update after every 10,000 training steps, which gives us a 1% replay rate.
- MBPA (Sprechmann et al., 2018): an episodic memory model that uses stored examples for local adaptation without sparse experience replay. The original MbPA formulation has a trainable key network. Our MbPA baseline uses a fixed key network since MbPA with a trainable key network performs significantly worse.
- MBPA$_{++}^{\mathrm{rand}}$: an episodic memory model with randomly retrieved examples for local adaptation (no key network).
- MBPA++: our episodic memory model described in §2.
- MTL: a multitask model trained on all datasets jointly, used as a performance upper bound.

### 4.3 Implementation Details

We use a pretrained BERT$_{\mathrm{BASE}}$ model (Devlin et al., 2018)[5] as our example encoder and key network. BERT$_{\mathrm{BASE}}$ has 12 Transformer layers, 12 self-attention heads, and 768 hidden dimensions (110M parameters in total). We use the default BERT vocabulary in our experiments.

We use Adam (Kingma & Ba, 2015) as our optimizer. We set dropout (Srivastava et al., 2014) to 0.1 and $\lambda$ in Eq. 1 to 0.001. We set the base learning rate to $3e^{-5}$ (based on preliminary experiments, in line with the suggested learning rate for using BERT). For text classification, we use a training batch of size 32. For question answering, the batch size is 8. The only hyperparameter that we tune is the local adaptation learning rate $\in \{5e^{-3}, 1e^{-3}\}$. We set the number of neighbors $K = 32$ and the number of local adaptation steps $L = 30$. We show results with other $K$ and sensitivity to $L$ in §4.5.

For each experiment, we use 4 Intel Skylake x86-64 CPUs at 2 GHz, 1 Nvidia Tesla V100 GPU, and 20 GB of RAM.

### 4.4 Results

The models are trained in one pass on concatenated training sets, and evaluated on the union of the test sets. To ensure robustness of models to training dataset orderings, we evaluate on four different orderings (chosen randomly) for each task. As the multitask model has no inherent dataset ordering, we report results on four different shufflings of combined training examples. We show the exact orderings in Appendix A. We tune the local adaptation learning rate using the first dataset ordering for each task and only run the best setting on the other orderings.

A main difference between these two tasks is that in text classification the model acquires knowledge about new classes as training progresses (i.e., only a subset of the classes that corresponds to a particular dataset are seen at each training interval), whereas in question answering the span predictor works similarly across datasets.

Table 1 provides a summary of our main results. We report (macro-averaged) accuracy for classification and $F_1$ score[6] for question answering. We provide complete per-dataset (non-averaged) results in Appendix B. Our results show that A-GEM outperforms the standard encoder-decoder model ENC-DEC, although it is worse than MBPA on both tasks. Local adaptation (MBPA) and sparse experience replay (REPLAY) help mitigate catastrophic forgetting compared to ENC-DEC, but a combination of them is needed to achieve the best performance (MBPA++).

Our experiments also show that retrieving relevant examples from memory is crucial to ensure that the local adaptation phase is useful. Comparing the results from MBPA++ and MBPA$_{++}^{\mathrm{rand}}$, we can

Table 1: Summary of results on text classification (above) and question answering (below) using averaged accuracy and $F_1$ score respectively (see Appendix A for the dataset orderings).

| Order | ENC-DEC | A-GEM | REPLAY | MBPA | MBPA$_{++}^{rand}$ | MBPA++ | MTL |
|---|---|---|---|---|---|---|---|
| i | 14.8 | 70.6 | 67.2 | 68.9 | 59.4 | **70.8** | 73.7 |
| ii | 27.8 | 65.9 | 64.7 | 68.9 | 58.7 | **70.9** | 73.2 |
| iii | 26.7 | 67.5 | 64.7 | 68.8 | 57.1 | **70.2** | 73.7 |
| iv | 4.5 | 63.6 | 44.6 | 68.7 | 57.4 | **70.7** | 73.7 |
| class.-avg. | 18.4 | 66.9 | 57.8 | 68.8 | 58.2 | **70.6** | 73.6 |
| i | 57.7 | 56.1 | 60.1 | 60.8 | 60.0 | **62.0** | 67.6 |
| ii | 55.1 | 58.4 | 60.3 | 60.1 | 60.0 | **62.4** | 67.9 |
| iii | 41.6 | 52.4 | 58.8 | 58.9 | 58.8 | **61.4** | 67.9 |
| iv | 58.2 | 57.9 | 59.8 | 61.5 | 59.8 | **62.4** | 67.8 |
| QA-avg. | 53.1 | 56.2 | 57.9 | 60.3 | 59.7 | **62.4** | 67.8 |

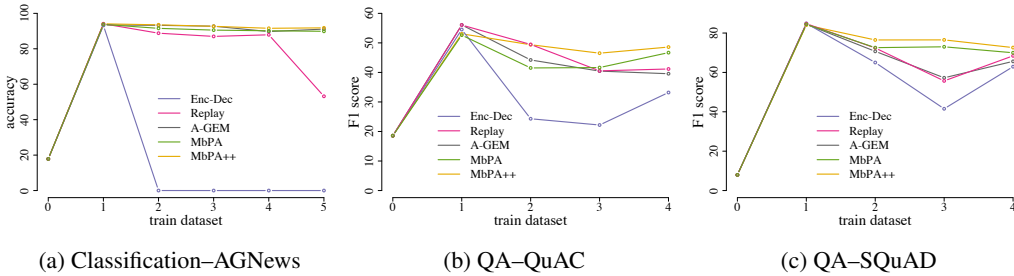

(a) Classification–AGNews      (b) QA–QuAC      (c) QA–SQuAD

Figure 2: Performance on test examples corresponding to the first dataset seen during training as training progresses.

see that the model that chooses neighbors randomly is significantly worse than the model that finds and uses similar examples for local adaptation. We emphasize that having a fixed key network is crucial to prevent representation drift. The original MBPA formulation that updates the key network during training results in a model that only performs slightly better than MBPA$_{++}^{rand}$ in our preliminary experiments. Our results suggest that our best model can be improved further by choosing relevant examples for sparse experience replay as well. We leave investigations of such methods to future work.

Comparing to the performance of the multitask model MTL—which is as an upper bound on achievable performance—we observe that there is still a gap between continual models and the multitask model.[7] MBPA++ has the smallest performance gap. For text classification, MBPA++ outperforms single-dataset models in terms of averaged performance (70.6 vs. 60.7), demonstrating the success of positive transfer. For question answering, MBPA++ still lags behind single dataset models (62.0 vs. 66.0). Note that the collection of single-dataset models have many more parameters since there is a different set of model parameters per dataset. See Appendix C for detailed results of multitask and single-dataset models.

Figure 2 shows $F_1$ score and accuracy of various models on the test set corresponding to the first dataset seen during training as the models are trained on more datasets. The figure illustrates how well each model retains its previously acquired knowledge as it learns new knowledge. We can see that MBPA++ is consistently better compared to other methods.

### 4.5 Analysis

**Memory capacity.** Our results in §4.4 assume that we can store all examples in memory (for all models, including the baselines). We investigate variants of MBPA++ that store only 50% and 10% of training examples. We randomly decide whether to write an example to memory or not (with probability 0.5 or 0.1). We show the results in Table 2. The results demonstrate that while the

Table 2: Results with limited memory capacity.

|        | 10%  | 50%  | 100% |
|--------|------|------|------|
| class. | 67.6 | 70.3 | 70.6 |
| QA     | 61.5 | 61.6 | 62.0 |

Table 3: Results for different # of retrieved examples $K$.

|        | 8    | 16   | 32   | 64   | 128  |
|--------|------|------|------|------|------|
| class. | 68.4 | 69.3 | 70.6 | 71.3 | 71.6 |
| QA     | 60.2 | 60.8 | 62.0 | -    | -    |

performance of the model degrades as the number of stored examples decreases, the model is still able to maintain a reasonably high performance even with only 10% memory capacity of the full model.

**Number of neighbors.** We investigate the effect of the number of retrieved examples for local adaptation to the performance of the model in Table 3. In both tasks, the model performs better as the number of neighbors increases.[8] Recall that the goal of the local adaptation phase is to shape the output distribution of a test example to peak around relevant classes (or spans) based on retrieved examples from the memory. As a result, it is reasonable for the performance of the model to increase with more neighbors (up to a limit) given a key network that can reliably compute similarities between the test example and stored examples in memory and a good adaptation method.

**Computational complexity.** Training MBPA++ takes as much time as training an encoder-decoder model without an episodic memory module since experience replay is performed sparsely (i.e., every 10,000 steps) with only 100 examples. This cost is negligible in practice and we observe no significant difference in terms of wall clock time to the vanilla encoder-decoder baseline. MBPA++ has a higher space complexity for storing seen examples, which could be controlled by limiting the memory capacity.

At inference time, MBPA++ requires a local adaptation phase and is thus slower than methods without local adaptation. This can be seen as a limitation of MBPA++ (and MBPA). One way to speed it up is to parallelize predictions across test examples, since each prediction is independent of others. We set the number of local adaptation steps $L = 30$ in our experiments. Figure 3 shows $L \approx 15$ is needed to converge to an optimal performance.

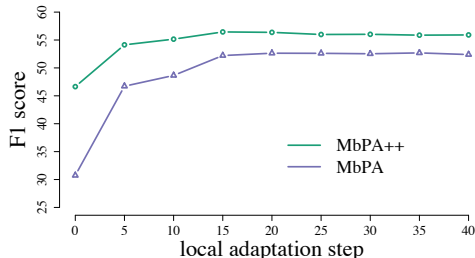

Figure 3: $F_1$ scores for MBPA++ and MBPA as the # of local adaptation steps increases.

Comparing MBPA++ to other episodic memory models, MBPA has roughly the same time and space complexity as MBPA++. A-GEM, on the other hand, is faster at prediction time (no local adaptation), although at training time it is slower due to extra projection steps and uses more memory since it needs to store two sets of gradients (one from the current batch, and one from samples from the memory). We find that this cost is not negligible when using a large encoder such as BERT.

**Analysis of retrieved examples.** In Appendix D, we show (i) examples of retrieved neighbors from our episodic memory model, (iii) examples where local adaptation helps, and (iii) examples that are difficult to retrieve. We observe that the model is able to retrieve examples that are both syntactically and semantically related to a given query derived from a test example, especially when the query is not too short and relevant examples in the memory are phrased in a similar way.

## 5 Conclusion

We introduced a lifelong language learning setup and presented an episodic memory model that performs sparse experience replay and local adaptation to continuously learn and reuse previously acquired knowledge. Our experiments demonstrate that our proposed method mitigates catastrophic forgetting and outperforms baseline methods on text classification and question answering.

**Acknowledgements**

We thank Gabor Melis and the three anonymous reviewers for helpful feedback on an earlier draft of this paper.

## Footnotes

[1]Contrast this with a more common setup where the model learns in a multitask setup (Ruder, 2017; McCann et al., 2018).

[2]McCann et al. (2018) show that many language processing tasks (e.g., classification, summarization, natural language inference, etc.) can be formulated as a question answering problem.

[3]Our preliminary experiments suggest that using only the question as the key slightly outperforms using the full input. Intuitively, given a question such as "Where was Barack Obama from?" and an article about Barack Obama, we would like to retrieve examples with similar questions rather than examples with articles about the same topic, which would be selected if we used the entire input (question and context) as the key.

[4] Future work can explore cheaper alternatives to gradient-based updates for local adaptation (e.g., a Hebbian update similar to the update that is used in plastic networks; Miconi et al., 2018).

[5]`https://github.com/google-research/bert`

[6]$F_1$ score is a standard question answering metric that measures $n$-grams overlap between the predicted answer and the ground truth.

[7] Performance on each dataset with the multitask model is better than or comparable to a single dataset model that is trained only on that dataset. Averaged performance of the multitask model across datasets on each task is also better than single-dataset models.

[8]We are not able to obtain results for question answering with $K = 64$ and $K = 128$ due to out of memory issue (since the input text for question answering can be very long).

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
