[Supplementary Material · supplementary.pdf]

# Appendices:
# Episodic Memory in Lifelong Language Learning

**Cyprien de Masson d'Autume, Sebastian Ruder, Lingpeng Kong, Dani Yogatama**
DeepMind
London, United Kingdom
{cyprien,ruder,lingpenk,dyogatama}@google.com

## A   Dataset Order

We use the following dataset orders (chosen randomly) for text classification:

   (i) Yelp → AGNews → DBPedia → Amazon → Yahoo.
  (ii) DBPedia → Yahoo → AGNews → Amazon → Yelp.
 (iii) Yelp → Yahoo → Amazon → DBpedia → AGNews.
 (iv) AGNews → Yelp → Amazon → Yahoo → DBpedia.

For question answering, the orders are:

   (i) QuAC → TrWeb → TrWik → SQuAD.
  (ii) SQuAD → TrWik → QuAC → TrWeb.
 (iii) TrWeb → TrWik → SQuAD → QuAC.
 (iv) TrWik → QuAC → TrWeb → SQuAD.

## B   Full Results

We show per-dataset breakdown of results in Table 1 (in the main paper) in Table 4 and Table 5 for text classification and question answering respectively.

## C   Single Dataset Models

We show results of a single dataset model that is only trained on a particular dataset in Table 6.

## D   Analysis of Retrieved Examples

We analyze retrieved examples to better understand how our model uses its episodic memory.

**Examples of retrieved neighbors.**   We show examples of retrieved neighbors from memory given a test query in Table 7. We observe that the model is generally able to retrieve relevant examples from the memory. In question answering, nearest neighbors tend to be examples that are both syntactically and semantically related. In text classification, they tend to be articles that discuss the a similar topic.

**Examples where local adaptation helps.**   In Table 8, we show two test examples where our model answers incorrectly before local adaptation, but correctly after. In the first case, we can see that training examples retrieved from memory are thematically related to the test example. In the second case, since the query is shorter, retrieved training examples tend to be more syntactically related.

Table 4: Per-dataset results on text classification for each ordering and model.

| Order | Model | Dataset | | | | |
|---|---|---|---|---|---|---|
| | | **1** | **2** | **3** | **4** | **5** |
| i | ENC-DEC | 1.1 | 0.0 | 0.0 | 4.3 | 68.7 |
| | A-GEM | 42.5 | 89.8 | 96.0 | 56.8 | 68.2 |
| | REPLAY | 38.2 | 83.9 | 95.4 | 50.3 | 67.9 |
| | MBPA | 42.0 | 90.4 | 96.1 | 52.0 | 63.9 |
| | $\text{MBPA}^{\text{rand}}_{++}$ | 35.2 | 80.4 | 88.2 | 45.9 | 47.2 |
| | MBPA++ | 45.7 | 91.6 | 96.3 | 54.6 | 65.6 |
| ii | ENC-DEC | 0.0 | 0.0 | 3.1 | 57.9 | 48.9 |
| | A-GEM | 80.1 | 50.3 | 91.3 | 57.3 | 50.6 |
| | REPLAY | 75.0 | 53.7 | 86.0 | 58.1 | 50.7 |
| | MBPA | 96.0 | 58.4 | 89.0 | 54.4 | 46.6 |
| | $\text{MBPA}^{\text{rand}}_{++}$ | 82.0 | 41.7 | 81.9 | 47.1 | 40.8 |
| | MBPA++ | 95.8 | 63.1 | 92.2 | 55.7 | 47.7 |
| iii | ENC-DEC | 0.0 | 0.0 | 1.3 | 11.4 | 93.9 |
| | A-GEM | 41.1 | 55.0 | 54.6 | 93.3 | 93.6 |
| | REPLAY | 23.6 | 36.8 | 25.0 | 94.5 | 93.8 |
| | MBPA | 43.3 | 60.9 | 51.6 | 95.8 | 92.5 |
| | $\text{MBPA}^{\text{rand}}_{++}$ | 35.2 | 33.6 | 42.1 | 92.3 | 82.3 |
| | MBPA++ | 44.3 | 62.7 | 54.4 | 96.2 | 93.4 |
| iv | ENC-DEC | 0.0 | 0.0 | 0.0 | 14.1 | 8.1 |
| | A-GEM | 90.8 | 44.9 | 60.2 | 65.4 | 56.9 |
| | REPLAY | 70.4 | 33.2 | 39.8 | 46.1 | 33.4 |
| | MBPA | 89.9 | 42.9 | 52.6 | 62.9 | 95.1 |
| | $\text{MBPA}^{\text{rand}}_{++}$ | 78.4 | 37.7 | 45.8 | 42.4 | 82.9 |
| | MBPA++ | 91.8 | 44.9 | 55.7 | 65.3 | 95.8 |

Although we only show the two nearest neighbors for each query here, our analysis provides an insight on ways our model uses its memory to improve predictions.

**Relevant examples that are difficult to retrieve.** In Table 9, we show two relevant training examples (as judged by humans) that are difficult to retrieve by the model (they are not in the 1,000 nearest neighbors) for the query `what was the name of bohemond s nephew`. The two relevant training examples ask about the nephew of a person, which is relevant for the given query. However, since they are phrased differently to the query, they are far in the embedding space, which is why a nearest neighbor method fails to retrieve these training examples. Our analysis shows that a better embedding and/or retrieval method can potentially improve the performance of our model.

Table 5: Per-dataset results on question answering for each ordering and model.

| Order | Model | Dataset | | | |
|---|---|---|---|---|---|
| | | **1** | **2** | **3** | **4** |
| i | ENC-DEC | 34.1 | 54.2 | 56.0 | 85.5 |
| | A-GEM | 36.7 | 51.8 | 53.4 | 82.5 |
| | REPLAY | 40.9 | 56.7 | 57.2 | 85.8 |
| | MBPA | 45.6 | 56.1 | 57.9 | 83.4 |
| | MBPA$_{++}^{rand}$ | 41.5 | 56.7 | 57.2 | 85.8 |
| | MBPA++ | 47.2 | 57.7 | 58.9 | 84.3 |
| ii | ENC-DEC | 61.9 | 64.2 | 29.3 | 65.0 |
| | A-GEM | 64.2 | 62.5 | 43.4 | 63.5 |
| | REPLAY | 67.0 | 64.1 | 44.9 | 65.2 |
| | MBPA | 69.9 | 63.4 | 43.6 | 63.3 |
| | MBPA$_{++}^{rand}$ | 67.5 | 62.5 | 46.5 | 63.7 |
| | MBPA++ | 72.6 | 63.4 | 50.5 | 63.0 |
| iii | ENC-DEC | 30.7 | 31.2 | 45.6 | 58.7 |
| | A-GEM | 47.6 | 47.0 | 57.4 | 57.4 |
| | REPLAY | 46.6 | 45.4 | 53.9 | 58.3 |
| | MBPA | 52.5 | 54.6 | 74.5 | 54.3 |
| | MBPA$_{++}^{rand}$ | 54.1 | 54.3 | 71.1 | 55.9 |
| | MBPA++ | 56.0 | 56.8 | 78.0 | 54.9 |
| iv | ENC-DEC | 55.5 | 37.1 | 54.8 | 85.4 |
| | A-GEM | 54.8 | 38.8 | 53.4 | 84.7 |
| | REPLAY | 56.9 | 41.8 | 56.4 | 86.1 |
| | MBPA | 58.0 | 47.2 | 57.4 | 83.3 |
| | MBPA$_{++}^{rand}$ | 55.5 | 43.0 | 54.6 | 85.9 |
| | MBPA++ | 59.0 | 48.7 | 58.1 | 83.6 |

Table 6: Performance of a standard encoder-decoder model on each dataset in our experiments. We report accuracy for text classification and $F_1$ score for question answering. We also show results from a multitask model for comparisons.

| Task | Dataset | Single Model | Multitask |
|---|---|---|---|
| Text Classification | AGNews | 93.8 | 94.0 |
| | Yelp | 50.7 | 50.3 |
| | Amazon | 60.1 | 58.8 |
| | Yahoo | 68.6 | 67.1 |
| | DBPedia | 30.5 | 95.9 |
| | Average | 60.7 | 73.2 |
| Question Answering | QuAC | 54.3 | 56.4 |
| | SQuAD | 86.1 | 85.7 |
| | Trivia Wikipedia | 62.3 | 64.0 |
| | Trivia Web | 62.4 | 64.4 |
| | Average | 66.0 | 67.6 |

Table 7: Examples of queries and retrieved nearest neighbors for question answering (top) and text classification (truncated, bottom). We also show the corresponding Euclidean distances in parentheses.

| **Query: in what country is normandy located** |
| --- |
| (17.48) in what area of france is calais located |
| (20.37) in what country is st john s located |
| (22.76) in what country is spoleto located |
| (23.12) in what part of africa is congo located |
| (23.83) on what island is palermo located |

| **Query: fears for t n pension after talks unions representing workers at turner newall say they are disappointed after talks with stricken parent firm federal mogul** |
| --- |
| (37.32) union anger at sa telecoms deal south african unions describe as disgraceful use of public money to buy telecoms shares for former government officials |
| (47.60) us hurting anti mine campaign anti landmine activists meeting in nairobi say us is setting bad example by not joining worldwide ban |
| (49.03) woolworths ordered to extend alh takeover deadline independent takeovers panel has headed off woolworths 39 attempts to force resolution in takeover battle for liquor retailer australian leisure and hospitality alh |
| (50.42) price hike for business broadband small net firms warn they could be hit hard by bt s decision to raise prices for business broadband |
| (51.08) job fears as deutsche culls chiefs deutsche bank is streamlining management of its investment banking arm raising fears that jobs may be lost in city german bank is reducing number of executives running its investment banking |

Table 8: Two examples where local adaptation helps.

| **Context:** david niven ( actor ) - pics , videos , dating , & news david niven male born mar 1 , 1910 james david graham niven , known professionally as david niven , was an english actor and novelist [. . . ] |
| --- |
| **Query:** in 1959 , for which film did david niven win his only academy award ? |
| **First two training examples retrieved from memory (2 nearest neighbors):** in which of her films did shirley temple sing animal crackers in my soup ? in 1968 , which american artist was shot and wounded by valerie solanis , an actress in one of his films ? |
| **Context:** dj kool herc developed the style that was the blueprint for hip hop music . herc used the record to focus on a short , heavily percussive part in it : the " break " . [. . . ] |
| **Query:** what was the break ? |
| **First two training examples retrieved from memory (2 nearest neighbors):** what was the result ? what was the aftermath ? |

Table 9: Relevant examples that are difficult to retrieve from memory.

| **Query:** what was the name of bohemond s nephew |
| --- |
| **Relevant examples not retrieved (Euclidean distances to the query in parentheses):** (87.88) who was the nephew of leopold (103.96) who is the nephew of buda king casimer iii the great |