[Reviews · NeurIPS 2019]

Reviewer 1



The paper is generally well written and clear. The contributions are substantial and represent an important step in applying pretrained models like BERT to continual learning setups. The submission appears to be technically sound, there is good analysis of the various components and complexity.

Reviewer 2



This paper proposes the neat idea of using a key-value store using pre-trained sequence representations as keys in order to train a model in a streaming fashion. The paper is very clear, experiments are convincing and this work will push this direction forward.

Reviewer 3



The paper addresses the very important topic of lifelong learning, and it proposes to employ an episodic memory to avoid catastrophic forgetting. The memory is based on a key-value representation that exploits an encoder-decoder architecture based on BERT. The training is made on the concatenation of different datasets, of which there is no need to specify the identifiers. The work is highly significant and the novelty of the contribution is remarkable. One point that would have deserved more attention is the strategies for the reading and writing of the episodic memory (see also comments below). Originality: high. Quality: high. Clarity: high. Significance: high.

[Author Response · NeurIPS 2019]

We thank the reviewers for their thoughtful comments.

**Reviewer 1**

a) We have conducted preliminary model dynamics and error analysis based on your feedback and summarize our results below. We will add them to an updated version of the paper and perform more analysis to improve it. This analysis was done on the question answering experiment.

**Examples where local adaptation helps.** In Table 1, we show two test examples where our model answers incorrectly before local adaptation, but correctly after. In the first case, we can see that training examples retrieved from memory are thematically related to the test example. In the second case, since the query is shorter, retrieved training examples tend to be more syntactically related. Although we only show the two nearest neighbors for each query here, our analysis provides an insight on ways our model uses its memory to improve predictions.

Table 1: Two examples where local adaptation helps.

| |
| --- |
| **Context:** david niven ( actor ) - pics , videos , dating , & news david niven male born mar 1 , 1910 james david graham niven , known professionally as david niven , was an english actor and novelist [. . .] |
| **Query:** in 1959 , for which film did david niven win his only academy award ? |
| **First two training examples retrieved from memory (2 nearest neighbors):** in which of her films did shirley temple sing animal crackers in my soup ? in 1968 , which american artist was shot and wounded by valerie solanis , an actress in one of his films ? |
| **Context:** dj kool herc developed the style that was the blueprint for hip hop music . herc used the record to focus on a short , heavily percussive part in it : the " break " . [. . .] |
| **Query:** what was the break ? |
| **First two training examples retrieved from memory (2 nearest neighbors):** what was the result ? what was the aftermath ? |

**Relevant examples that are difficult to retrieve.** In Table 2, we show two relevant training examples (as judged by humans) that are difficult to retrieve by the model (they are not in the 1,000 nearest neighbors) for the query `what was the name of bohemond s nephew`. The two relevant training examples ask about the nephew of a person, which is relevant for the given query. However, since they are phrased differently to the query, they are far in the embedding space, which is why a nearest neighbor method fails to retrieve these training examples. Our analysis shows that a better embedding and/or retrieval method can potentially improve the performance of our model.

Table 2: Relevant examples that are difficult to retrieve from memory.

| |
| --- |
| **Query:** what was the name of bohemond s nephew |
| **Relevant examples not retrieved (Euclidean distances to the query in parentheses):** (87.88) who was the nephew of leopold (103.96) who is the nephew of buda king casimer iii the great |

b) We will reorganize the presentation of the main results (Table 1 in the paper) to include some results from Appendix B such that they are more informative for readers who work with datasets we consider in our paper.

c) We will revise the characterization of our work with respect to McCann et al. according to your suggestion.

**Reviewer 2**

a) Examples from each dataset need not be seen contiguously for our model to work. In the limit, when all examples across datasets are shuffled, we reach the performance of the multitask upper bound shown in the paper.

b) For classification, yes, it is correct that we have a 33-way softmax. These 33 classes already include overlapping classes from two datasets (Yelp and Amazon).

**Reviewer 3**

a) Thank you for your suggestions and pointers to typos. We will add more details about our experiments with other reading and writing strategies as an appendix.

[Meta-Review · NeurIPS 2019]

This paper proposes the use of memory in life-long learning to prevent catastrophic forgetting by means of experience replay and local adaptation. The idea is simple yet it is an interesting new step in this line of work. The experiments are comprehensive. The writing is clear. The paper would be a good addition to the conference, and has support from reviewers.